# Photocatalytic Hydrogen Evolution Using Bi-Metallic (Ni/Pt) Na$_2$Ti$_3$O$_7$ Whiskers: Effect of the Deposition Order

**Luis F. Garay-Rodríguez** [1], **S. Murcia-López** [2,*], **T. Andreu** [2], **Edgar Moctezuma** [3], **Leticia M. Torres-Martínez** [1,*] **and J. R. Morante** [2]

[1] Facultad de Ingeniería Civil-Departamento de Ecomateriales y Energía, Universidad Autónoma de Nuevo León, Cd. Universitaria, 66455 San Nicolás de los Garza, N.L., Mexico; lfgarayr@gmail.com

[2] Catalonia Institute for Energy Research (IREC), Jardins de les Dones de Negre 1, 08930 Sant Adrià de Besós, Spain; tandreu@irec.cat (T.A.); jrmorante@irec.cat (J.R.M.)

[3] Facultad de Ciencias Químicas, Universidad Autónoma de San Luis Potosí, Av. Manuel Nava #6, 78290 San Luis Potosí, S.L.P., Mexico; edgar@uaslp.mx

\* Correspondence: smurcia@irec.cat (S.M.-L.); lettorresg@yahoo.com (L.M.T.-M.); Tel.: +34-93-356-26-15 (S.M.-L.); +52-(81)-14-42-44-00 (ext. 7293) (L.M.T.-M.)

**Abstract:** Photocatalytic hydrogen production through ethanol photo-reforming using Na$_2$Ti$_3$O$_7$ whiskers increases if the sodium titanate is decorated with well-known metallic catalysts such as Ni and Pt. Whereas wet impregnation with nickel gives only a slight increase in the activity, photo-deposition of Pt increased the H$_2$ production by more than one order of magnitude. Through the combination of both co-catalysts (Ni and Pt) a superior performance in terms of H$_2$ production is further observed. However, hydrogen yield is largely enhanced (almost three-fold), up to 778 μmol·g$^{-1}$·h$^{-1}$, if the Pt is photo-deposited on the surface of the catalyst before wet impregnation with Ni species (NTO/Pt/Ni) compared to H$_2$ yield (283 μmol·g$^{-1}$·h$^{-1}$) achieved with the catalyst prepared in the reverse order (NTO/Ni/Pt). Structural, morphological, optical, and chemical characterization was carried out in order to correlate physicochemical properties with their photocatalytic activity. The X-ray photoelectron spectroscopy (XPS) results show a higher concentration of Pt$^{2+}$ species if this metallic layer is under the nickel oxide layer. Moreover, X-ray diffraction patterns (XRD) show that Na$_2$Ti$_3$O$_7$ surface is modified for both metal decoration processes.

**Keywords:** Na$_2$Ti$_3$O$_7$; metal deposition; photo-reforming

## 1. Introduction

In recent years, hydrogen has attracted a lot of attention as one of the most promising substitute of fossil fuels, as a clean energy carrier because its calorific power per mass unit (142 MJ/kg) which is higher than those of such fuels as CH$_4$ (55.5 MJ/kg), carbon (~30 MJ/kg), and gasoline (46.4 MJ/kg) [1,2]. Some strategies have been studied for its production, the heterogeneous photocatalysis being one alternative of great interest considering the possibility of obtaining H$_2$ from water or organic compounds, such as alcohols, by simply using solar radiation. In this sense, photo-reforming of organic molecules, such as alcohols, for hydrogen evolution in both liquid and gas phase, has attracted a lot of attention recently mainly because the organic molecules act as hole scavengers and undergo relatively fast and irreversible oxidation [3].

Among several semiconductor metal oxides, TiO$_2$ has been one of the most reported photocatalysts for degradation of organic molecules, hydrogen production by the water splitting reaction and CO$_2$

reduction under UV and visible light irradiation [4–7]. To overcome the $TiO_2$-based system limitations, several strategies have been developed, including the use of co-catalysts to accelerate the charge transfer and avoid recombination or the modification of the semiconductor chemical or crystalline structure to improve the light absorption processes.

Regarding the use of co-catalysts, Ni has been one of the most reported metallic oxides to improve photocatalytic activity of $TiO_2$ for hydrogen production [8,9]. Some recent studies have also shown an enhanced photocatalytic activity of semiconductor catalysts partially covered or a bi-metallic catalyst which gives rise to an improved charge transfer rate [10,11].

In addition to $TiO_2$, materials based on the family of inorganic alkali titanates (with the general formula of $A_2Ti_nO_{2n+1}$) have also shown to be promising alternatives for hydrogen generation through photocatalytic water splitting reactions without sacrificial agents. These materials possess a structure consisting of edge and corner $TiO_6$ octahedral in units of $(Ti_3O_7)^{2-}$ layers held together by different ions forming layered tunnels [12]. More importantly, their higher efficiencies in the photo-induced processes [13,14], lithium batteries [15,16], chemical adsorption [17,18], and biomedics [19,20], have been attributed to this crystalline arrangement. Some of the most reported alkali titanates based on Na, K, and Rb have been evaluated in the photocatalytic hydrogen evolution reaction under UV–Vis irradiation, being the Na titanates the most efficient photocatalysts under different reaction conditions. This is mainly attributed to the lower distortion of its crystalline structure as a result of the use of a cation with a lower atomic radius [21,22].

Therefore, sodium tri-titanate ($Na_2Ti_3O_7$) has been prepared by different synthesis methods, leading to nanostructures with varied morphologies such as particles, rods, tubes, belts, and ribbons [23,24], which have an important effect on their properties for specific applications. For instance, $Na_2Ti_3O_7$ belts have shown high activity for the photocatalytic degradation of organic pollutants and hydrogen evolution in liquid phase batch reactors. In particular, the superior photocatalytic activity of sodium titanates doped with hydrogen or impregnated with different co-catalysts, such as noble metals (Ag, Au, Pt) or $In_2S_3$, has been attributed to the low charge recombination rate and to the capacity of absorbing more energy under UV–Vis illumination [25,26]. In fact, despite the higher band gap of $Na_2Ti_3O_7$ (3.6 eV) [26], the conduction band potential is thermodynamically negative enough for a better performance in the hydrogen evolution reaction [27].

In this scenario, this work presents the evaluation of the photocatalytic activity of $Na_2Ti_3O_7$ whiskers prepared by solid-state reaction in the ethanol photo-reforming process to produce hydrogen in a continuous gas phase reactor under UV radiation. For this purpose, special interest has been paid on the improvement achieved by the use of nickel and platinum co-catalysts. Therefore, titanate surface was decorated with Ni by the wet impregnation method and with Pt by photo-deposition in order to obtain nickel oxide and metallic platinum species [28,29] that will modify the photocatalytic properties of the semiconductor.

## 2. Results and Discussion

### 2.1. Characterization

Figure 1a presents the XRD patterns of the bare $Na_2Ti_3O_7$ (NTO) and catalysts loaded with Ni or Pt (NTO/Ni and NTO/Pt), where sodium titanate was the main observed phase. Due to the low concentration of co-catalysts, there is an absence of Pt or Ni species reflections in the patterns.

The diffractogram of the nickel loaded catalyst shows some extra small spikes in the baseline at around $10.9°$ and $24.2°$; these reflections are more evident on the diffractogram of the platinum-loaded catalyst. These extra reflections correspond to some of the main reflections of the $H_2Ti_3O_7$ phase (JCPDS 00-036-0654), suggesting the protonation of the synthesized $Na_2Ti_3O_7$ as a result of impregnation (Ni) or photo-deposition (Pt) conditions. Furthermore, the Pt-loaded sample presents very small reflections around $14.5°$ and $18.4°$ which correspond to the $NaTi_8O_{13}$ phase; however, as a result of their imperceptible intensity, they can be omitted. The majority phase in the patterns ($Na_2Ti_3O_7$) in

Figure 1a displays a monoclinic crystalline structure (JCPDS 00-031-1329) with an average crystallite size, calculated by the Scherrer equation using the (0 0 1) reflection of around 75 nm.

More complex features were observed in the bimetallic samples as shown in the XRD patterns in Figure 1b. As it can be noticed, both samples present more extra reflections with higher intensities than the single metal loaded ones, which are representative of a modification of the pristine phases. They may be due to Na deficiency in the crystalline structure of sodium titanate and these reflections may indicate the formation of $NaTi_8O_{13}$ and $H_2Ti_3O_7$ phases (JCPDS 00-048-0523 and 00-036-0654, respectively), being the first one representative of a family of sodium titanates with $Ti^{3+}/Ti^{4+}$ mixed-valence [30]. In addition, it is evident that the (0 0 1) reflection moved to slightly higher 2θ values, which also suggests the partial replacement of sodium atoms to form $Na_{2-x}H_xTi_3O_7$. According to some studies, these phases are intermediate products of the transformation of $Na_2Ti_3O_7$ to $Na_2Ti_6O_{13}$ under extreme conditions [31]. In this context, we can assume that the bi-metallic deposition on the surface of sodium titanate under the performed reaction conditions favors the partial reduction of titanium and the substitution of the sodium atoms by hydrogen atoms.

It is evident that the intensity of the (0 0 1) reflection ($\approx$10.5°) is different comparing bare and bi-metallic samples. The most noticeable decrease in this pattern is presented by NTO/Ni/Pt; however, NTO/Pt/Ni is also less intense compared to the bare NTO (Figure S1). This reduction in the (0 0 1) titanate main intensity is related with a decrease in the interlayer distance between $TiO_6$ octahedrons [32] suggesting a major replacement of Na by smaller cations such as Pt, Ni or H in the NTO/Ni/Pt sample and reducing the interlayer distance; furthermore, this sample presents more intense peaks of the main $H_2Ti_3O_7$ phase (24°, 28° and 48°) compared to the NTO/Pt/Ni, corroborating this interlayer reduction and suggesting the major presence of this phase instead of the $NaTi_8O_{13}$. A contrary behavior is observed in NTO/Pt/Ni sample, where $H_2Ti_3O_7$ phase peaks are less intense compared to the $NaTi_8O_{13}$ (17.8° and 27.8°), suggesting the higher presence of the $Ti^{3+}/Ti^{4+}$ mixed-valence on this catalyst.

A less marked behavior is observed in single metal-loaded samples, as it is observed, also in Figure S1, Ni- and Pt-loaded NTO present a reduction in the (0 0 1) reflection intensity, as a result of the decrease in the interlayer distance and more evident in the NTO/Ni sample, which suggests the possible introduction of Ni species between the $TiO_6$ octahedrons.

In this context, it is important to highlight that the metal deposition order produces a different development in the formed faces from NTO.

Bi-metallic photocatalysts were analyzed by XPS to obtain more information about their chemical composition.

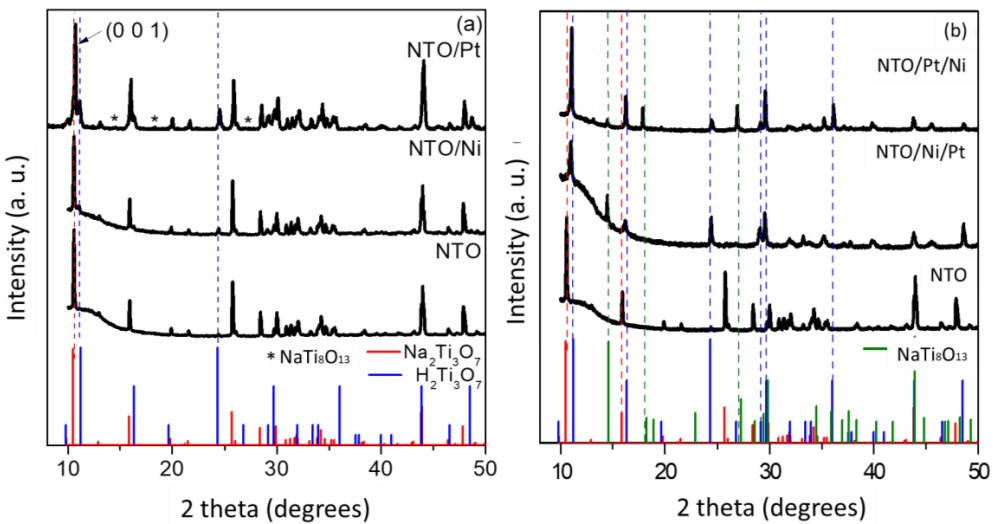

**Figure 1.** XRD patterns of $Na_2Ti_3O_7$ pure and loaded with Ni and Pt (**a**), and NTO/Ni/Pt and NTO/Pt/Ni (**b**).

Figure S2 shows the XPS survey of NTO, NTO/Ni/Pt, and NTO/Pt/Ni samples. As it is evident, some peaks come from the covered substrate, Na and Ti, while others come from the deposited metals (Pt and Ni) being much more intense the Pt signal in NTO/Ni/Pt sample compared to the NTO/Pt/Ni one, whereas the Ni signal is less intense. Carbon corresponds to some surface contamination whereas the signal of oxygen comes from the NTO substrate, possible oxygen bond with the surface carbon contamination, as well as from the oxidized state of the Ni and Pt.

Figure 2 shows the de-convoluted spectra of Na 1s, Ti 2p, and O 1s species in the bare $Na_2Ti_3O_7$ sample; as it can be noticed, Ti 2p spectrum was resolved in three single peaks characteristic of $Ti2p_{3/2}$ and $Ti 2p_{1/2}$ orbitals and their respective satellite, all of them located at 458.1, 463.8, and 471.3 eV, respectively, and representative of $Ti^{4+}$ species [33]. On the contrary, Na 1s and O 1s present both a single peak located at 1071.1 and 529.7 eV, respectively, which are related to the presence of $Na^+$ and $O^{2-}$ [34,35] species in the sample. All of these peaks corroborates the presence of a single $Na_2Ti_3O_7$ phase.

A different behavior is observed in the NTO/Ni/Pt sample, which is evidenced on its de-convoluted XPS spectra presented in Figure 3. In this context, the Ti 2p region was resolved on its representative Ti $2p_{3/2}$ and $2p_{1/2}$ peaks and its satellite at 472.8 eV. Both Ti $2p_{3/2}$ and $2p_{1/2}$ peaks were fitted in two doublet ones centered at 459, 460, 464, and 465.7 eV, respectively. Both peaks located at 459 and 465.7 are representative of the $Ti^{4+}$ species [33]; meanwhile, the 460 and 464 peaks correspond to the $Ti^{3+}$ species [36]. The presence of these peaks corroborates the $Ti^{4+}/Ti^{3+}$ mixture in the intermediate phase formed during bi-metallic synthesis ($NaTi_8O_{13}$). A peak of Na 1s was found at 1071.95 eV (Figure 3) as evidence of $Na^+$ species present in the sample [34]. The O 1s spectrum was resolved in a very intensive peak at 530 eV, corresponding to $O^{2-}$ species [35] related to the formation of O–M bonds, where M can be any of the present metals. A peak at 534 eV was resolved suggesting the presence of water in the surface, as a result of the lack of thermal treatment after Pt photo-deposition process. This step was omitted in order to avoid the Pt oxidation by temperature.

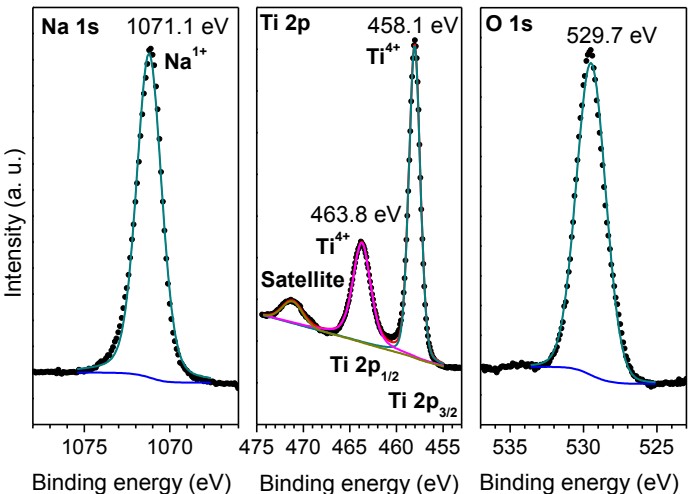

**Figure 2.** XPS spectra of Na 1s, Ti 2p, and O 1s in the NTO sample.

Ni 2p and Pt 4f spectra are also shown in Figure 3. The Ni 2p spectrum was resolved in characteristic Ni $2p_{3/2}$ and $2p_{1/2}$ peaks, located at around 855.5 and 873 eV; furthermore, the respective satellites of these peaks were found at around 860.5 and 880 eV. These signals are associated to $Ni^{2+}$ oxidation state, [37] which might exist in the form of $Ni(OH)_2$ over the titanate surface, as expected by the impregnation conditions (calcination at 400 °C after impregnation and further Pt photo-deposition without calcination). On the other hand, the Pt 4f spectrum was de-convoluted into four peaks, with the most intense one located at 76.1 eV, characteristic of Pt $4f_{7/2}$ and attributed to $Pt^{4+}$ species. A second peak was resolved at 79.8 eV, which is also associated with $Pt^{4+}$ (Pt $4f_{5/2}$) [38]. Other peaks with lower intensities were resolved at 74.1 and 77.2 eV associated with Pt $4f_{7/2}$ and Pt $4f_{5/2}$ ($Pt^{2+}$).

The very high intensity of $Pt^{4+}/Pt^{2+}$ peaks is indicative of partially oxidized states as a consequence of an incomplete photo-deposition process, which has also been reported for some photo-deposition methods under certain conditions [39]. The XPS quantitative analyses of this sample for each element peak are collected in Table S1.

The XPS survey spectrum of NTO/Pt/Ni sample is presented also in Figure S2. As in the NTO/Ni/Pt, representative peaks of the expected elements were observed. The de-convoluted spectra obtained from the NTO/Pt/Ni sample are shown in Figure 4. In this case, as Ni was impregnated in the outer layer, the intensities of its peaks are relatively higher than those of Pt. In addition, Na 1s, Ti 2p, and O 1s peaks were found also at similar binding energies than the previous sample, thus confirming that these elements are in the same oxidation states. In addition, Ti 2p spectra also presented the characteristic peak of $Ti^{3+}$ species conforming again the presence of a mixture of $Ti^{4+}/Ti^{3+}$ in the intermediate phases formed. Compared to the NTO/Ni/Pt sample, the O 1s spectrum mainly corresponds to the O–M bonds. The Ni 2p spectrum was de-convoluted with the Ni $2p_{3/2}$ and $2p_{1/2}$ signals located at around 855.39 and 873.02 eV with their respective satellites at around 861.11 and 879.36 eV, which are associated with $Ni^{2+}$ oxidation state. With this information, we can conclude that the co-catalyst deposition order does not affect the oxidation state of Ni because the catalysts are calcined under air atmosphere and the metallic particles impregnated over titanate are apparently oxidized to $Ni^{2+}$ species (NiO or $Ni(OH)_2$) [37]. On the other hand, the Pt 4f spectra, unlike the previous sample, now present an oxidation state $Pt^{2+}$ as a predominant platinum contribution. The peaks found at 73.6 and 76.5 eV are characteristic of Pt $4f_{5/2}$ and Pt $4f_{7/2}$ attributed to $Pt^{2+}$ species. Furthermore, the other peaks at 75.5 and 79.5 eV are representative of Pt $4f_{7/2}$ and Pt $4f_{5/2}$ ($Pt^{4+}$) species. This mayor difference in oxidation states could suggest the presence of several phases such as $PtO_2$, or PtO species under the photo-deposition. Likewise, the posterior nickel impregnation seems to prevent full oxidation of the platinum.

Table 1 shows the calculated ratios between $Ti^{4+}/Na$, $Ti^{3+}/Ti^{4+}$, and $Pt^{2+}/Pt^{4+}$ from both samples. The slightly higher value of $Ti^{3+}/Ti^{4+}$ ratio in NTO/Pt/Ni sample suggests the higher number of oxygen vacancies in this sample compared to NTO/Ni/Pt (0.46 vs. 0.36) as a result of the formation of the intermediate phase $NaTi_8O_{13}$; this information confirms the results obtained by XRD, with more intense peaks associated with this phase on this sample. The lower value of the $Ti^{4+}/Na$ ratio in the NTO/Ni/Pt sample gives information about the higher Na loss on this sample because of the possible Ni substitution between the $TiO_6$ octahedrons or its fast transition to the $H_2Ti_3O_7$ phase. This also agrees with the XRD patterns, as the NTO/Ni/Pt sample presents more intense diffraction peaks corresponding to the $H_2Ti_3O_7$ protonated phase compared to the other one (NTO/Ni/Pt). Finally, the higher $Pt^{2+}/Pt^{4+}$ ratio on the NTO/Pt/Ni sample (1.45 vs. 0.32) suggests a higher Pt reduction efficiency than over bare NTO and over Ni-loaded NTO, which can be related to the phase transition to the mixed valence phase and the formation of oxygen vacancies.

The C 1s spectra of both materials also present differences in the surface contamination. Due to the Pt photo-deposition after Ni in the NTO/Ni/Pt sample (Figure 3), an extra peak at 286.3 eV is observed; this peak could suggest the presence of carbonates on the catalyst surface (as confirmed also by the 531.7 eV peak in O 1s spectrum) associated to the use of isopropanol during the photo-deposition reaction which was not completely removed by washing. However, both peaks disappear completely in the NTO/Pt/Ni sample as a result of thermal treatment at 400 °C, carried out after Ni impregnation.

**Table 1.** XPS ratios obtained from NTO/Ni/Pt and NTO/Pt/Ni samples.

| Sample | $Ti^{4+}/Na$ | $Ti^{3+}/Ti^{4+}$ | $Pt^{2+}/Pt^{4+}$ |
|---|---|---|---|
| NTO/Ni/Pt | 0.78 | 0.36 | 0.32 |
| NTO/Pt/Ni | 0.28 | 0.46 | 1.45 |

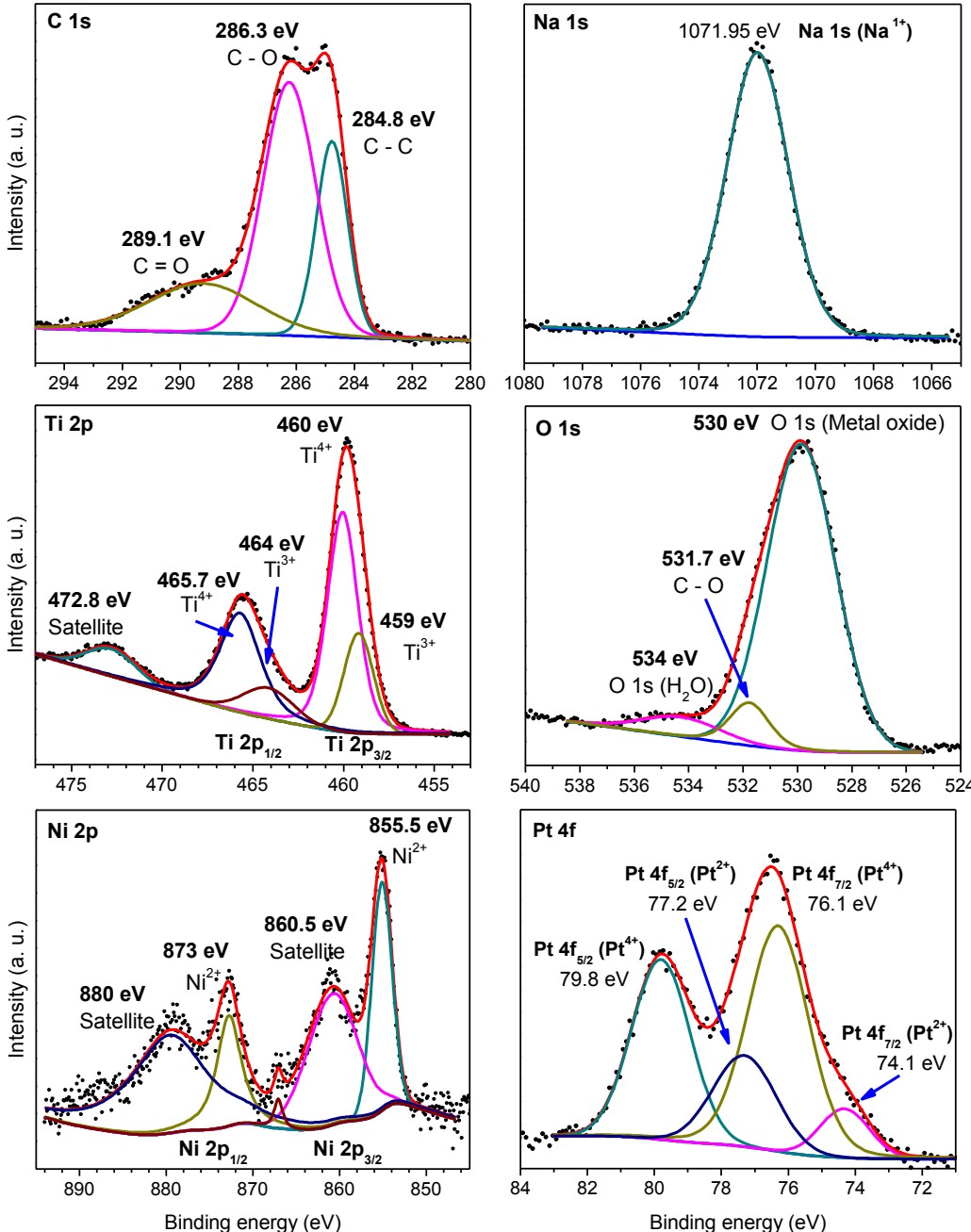

**Figure 3.** XPS spectra in C 1s, Ti 2p, Na 1s, O 1s, Ni 2p, and Pt 4f zones of the NTO/Ni/Pt sample.

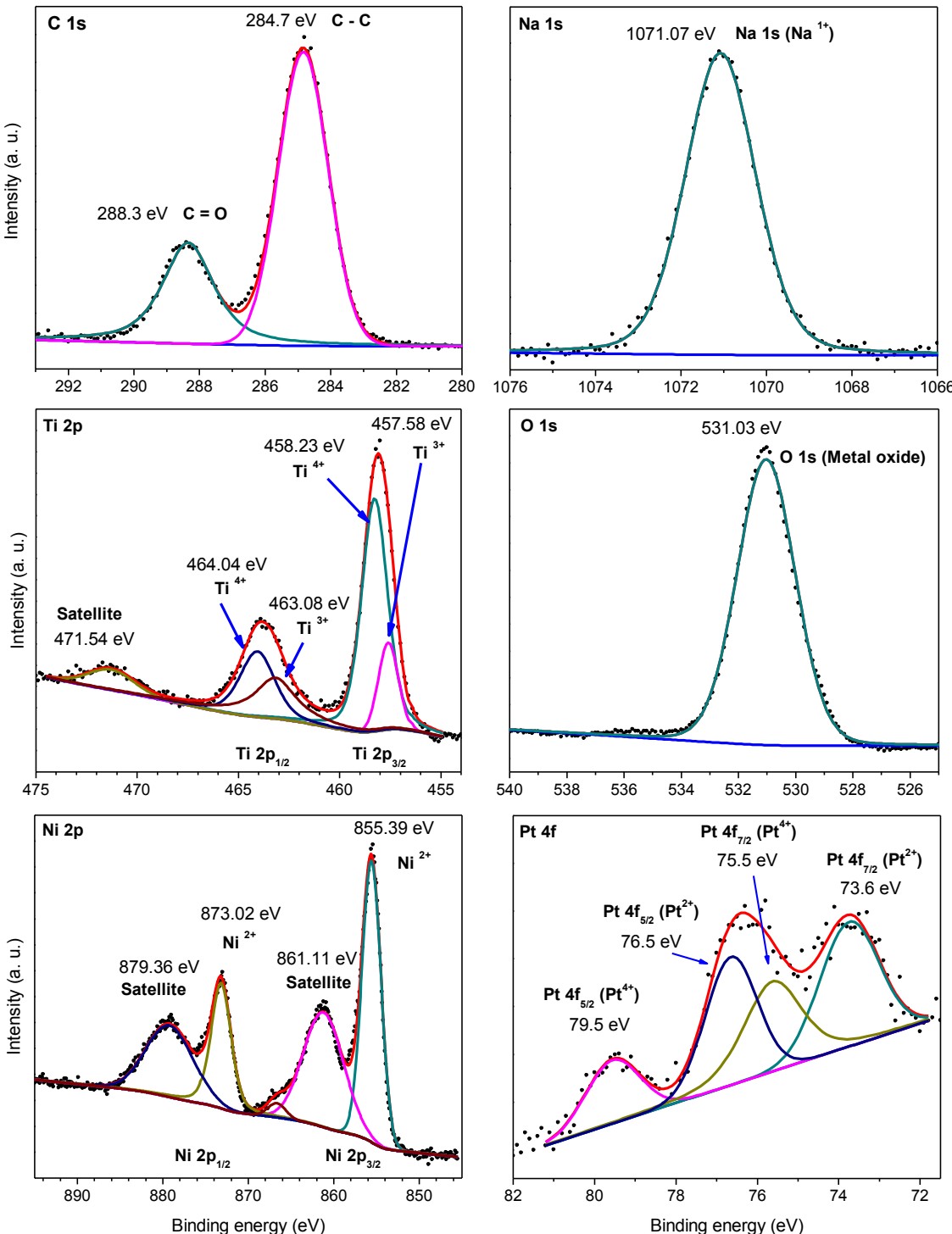

**Figure 4.** XPS spectra in C 1s, Ti 2p, Na 1s, O 1s, Ni 2p, and Pt 4f zones of the NTO/Pt/Ni sample.

SEM images of $Na_2Ti_3O_7$ samples are shown in Figure 5a,b. As it can be seen, this material exhibits a whisker-like morphology, which is characteristic of some alkali metal titanates [40]. In general, the samples have an apparent average whisker length of around 5 μm, resulting from the high temperatures used during solid-state reaction synthesis.

Micrographs of the samples loaded with Ni/Pt and Pt/Ni are shown in Figure 5c,d, respectively. It is evident that metal impregnation or photo-deposition did not modify the morphology of the support. EDS analyses confirmed the presence of Ni and Pt in both samples. The quantitative EDS

analyses are presented in Table 2. As it can be noticed, the Na loss during both co-catalyst depositions as a result of the transition to the protonated phase is evident. More notorious changes are presented when bi-metallic samples are evaluated; in concordance to XRD and XPS analysis, more Na deficiency was observed in the NTO/Pt/Ni sample compared to NTO/Ni/Pt, suggesting the transition to the mixed valence phase and the protonated one. Pt and Ni as first co-catalyst (samples NTO/Ni and NTO/Pt) seem not to show a significant change even with the second metal deposition process; for that reason, their concentration was similar in NTO/Ni/Pt and NTO/Pt/Ni.

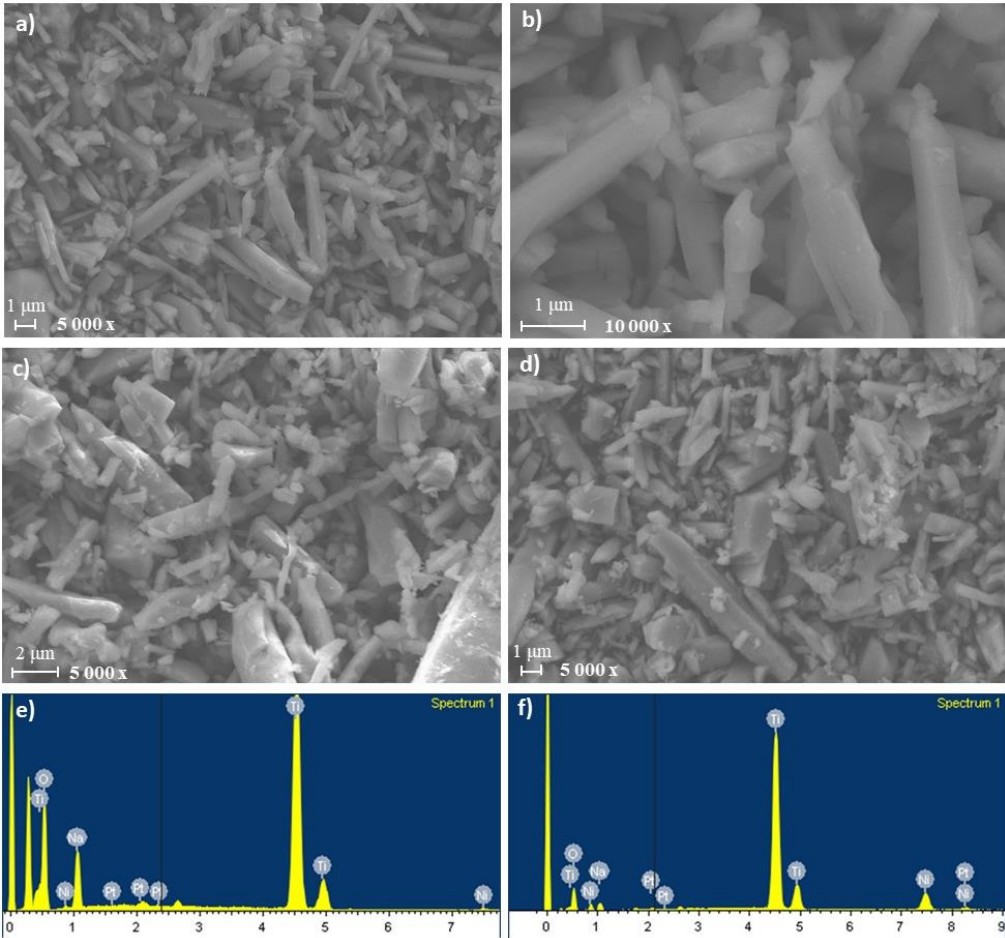

**Figure 5.** SEM images of NTO (**a**,**b**), NTO/Ni/Pt (**c**), and NTO/Pt/Ni (**d**), and EDX analysis of NTO/Ni/Pt (**e**) and NTO/Pt/Ni (**f**).

**Table 2.** Elemental quantification of the synthesized catalysts.

| Material | Atomic (%) Quantification | | | Weight (%) Quantification | | |
|---|---|---|---|---|---|---|
| | Na | Pt | Ni | Na | Pt | Ni |
| NTO | 8.5 | – | – | 9.2 | – | – |
| NTO/Ni | 8.4 | – | 0.7 | 8.9 | – | 1.2 |
| NTO/Pt | 8.1 | 0.1 | – | 8.4 | 0.8 | – |
| NT/Ni/Pt | 7.4 | 0.2 | 0.5 | 7.9 | 1.0 | 1.1 |
| NT/Pt/Ni | 3.4 | 0.1 | 0.9 | 4.8 | 0.7 | 1.0 |

$N_2$ adsorption was used for determining the BET surface area of the samples. Due to the high-temperature thermal treatment applied during the synthesis method, the bare titanate exhibits a relatively low surface area (<10 $m^2 \cdot g^{-1}$), which practically remains unmodified after single Ni or Pt decoration. However, in the case of bimetallic samples, surface areas had a significate variation,

which can be attributed to the secondary process. The NTO/Ni/Pt sample exhibited a BET area increase, although the value remains lower than 10 $m^2 \cdot g^{-1}$. On the contrary, the NTO/Pt/Ni sample presented a six-fold increase (28 $m^2 \cdot g^{-1}$) compared with sample only loaded with Pt (4 $m^2 \cdot g^{-1}$). In addition, pore size distributions of both samples are presented in Figure 6. As it is evident, higher pore volume was obtained in sample NTO/Pt/Ni (0.029 $cm^3 \cdot g^{-1}$) compared with NTO/Ni/Pt (0.009 $cm^3 \cdot g^{-1}$). This difference in pore volume between samples can be attributed to the sodium loss during the Pt photo-deposition and Ni impregnation processes forming some intermediate phases with higher pore volume than bare titanate. The results from BET analysis of all materials synthesized are summarized in Table 3.

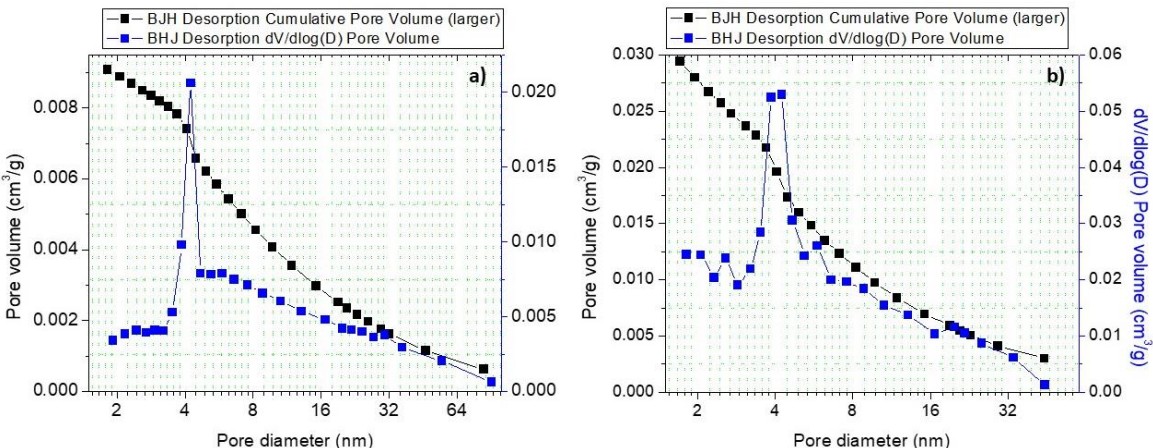

**Figure 6.** Pore size distribution of (**a**) NTO/Ni/Pt and (**b**) NTO/Pt/Ni samples.

In order to perform the optical characterization, diffuse reflectance UV–VIS analysis of the samples was carried out. Figure 7 shows the diffuse reflectance spectra of NTO metallic and bi-metallic samples. A sharp decrease in reflectance is observed around 365 nm, which makes evident the higher absorption of these materials under UV radiation. As it is evident, a decrease in reflectance is observed with the Ni impregnation. In addition, the NTO/Pt/Ni sample presents a transition around of 550 nm, which is representative of $Ni(OH)_2$ species [41] confirming the previous assumption made with XPS results. On the contrary, this transition is less visible on the NTO/Ni/Pt sample, probably due to the Pt species in the outer layer. No $Ni(OH)_2$ transitions were observed in the NTO/Ni sample as a result of the thermal treatment performed after impregnation suggesting the presence of only NiO particles. Furthermore, some other transitions less marked are observed, mainly in the Pt loaded samples between 500–800 nm; some of them can be considered representative of $PtO_2$ or $Pt(OH)_2$ species [42,43], confirming the presence of these phases as a result of the incomplete Pt reduction over catalysts.

With this data and by using the Kubelka–Munk transformation, Tauc plots were obtained, as shown in Figure S3. As seen, a minor variation in the bandgap values (Table 3) is observed after single metal or bi-metallic deposition. In this context, the bare NTO presents a bandgap value of 3.4 eV, which is similar to the previously reported in the literature [26]. The lowest value was calculated for the NTO/Ni sample (3.25 eV); this reduction in Eg can be related to the presence of $Ni^{2+}$ species between $TiO_6$ octahedrons as a result of its exchange during the impregnation conditions, creating impurity states within the band gap and resulting in a reduction of itself [25,44].

In general, no significant variation is found in the modified samples as a result of the very low concentration of loaded metals (1 wt %).

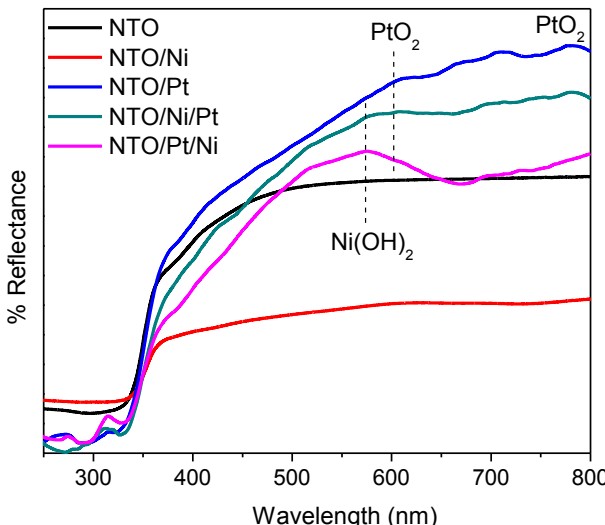

**Figure 7.** Diffuse reflectance spectra of $Na_2Ti_3O_7$ bare and deposited with Ni and Pt metals.

**Table 3.** Textural and optical characterization of the synthesized catalysts.

| Surface Area ($m^2$/g) | Pore Volume ($cm^3$/g) | Bandgap (eV) |
|---|---|---|
| 2 | 0.002 | 3.40 |
| 3 | 0.003 | 3.25 |
| 4 | 0.004 | 3.43 |
| 7 | 0.009 | 3.45 |
| 28 | 0.029 | 3.36 |

As a result of the further characterization of the prepared catalysts, it is possible to propose a reaction mechanism of the phase transition during both processes (impregnation and photo-deposition). Figure 8 shows the proposed mechanism of the deposition of a single co-catalyst. As it can be observed, during the Ni impregnation, as a result of the performed conditions, it is possible to exchange some $Na^+$ species by $Ni^{2+}$ ones into the $TiO_6$ layers [45], reducing the interlayer distance between them. Furthermore, NiO species were deposited over the NTO surface.

A different behavior was observed as a result of the Pt photo-deposition conditions. In this context, thanks to the catalyst illumination, electrons ($e^-$) and holes ($h^+$) are produced in the NTO surface. These species are responsible for the isopropanol degradation ($h^+$) and Pt precursor reduction ($e^-$) [39,46]; however, electrons are also capable of the $Ti^{4+}$ species reduction producing $Ti^{3+}$ ones [47]. This reduction is responsible for the oxygen vacancies generation and the formation of the $NaTi_8O_{13}$ phase ($Ti^{3+}$/$Ti^{4+}$ mixed valence). From the isopropanol degradation, protons ($H^+$) are generated and this species can be exchanged by Na into the layered tunnels resulting in the formation of the protonated phase ($Na_{2-x}H_xTi_3O_7$).

A similar development is presented in the case of bi-metallic samples. In this context, it is important to remind that the starting material for the second deposition is the obtained from the single metal loaded catalyst explained previously. The proposed mechanism for the synthesis of the bi-metallic catalyst is presented in Figure 9.

As it can be observed, for the Pt photo-deposition in the second layer (NTO/Ni/Pt sample), a $Na^+$ exchanged by $Ni^{2+}$ sample is photo-activated. In this context, as a result of this exchange, a favored transition to the $H_2Ti_3O_7$ phase is observed, probably associated to a higher generation of protons. Unfortunately, electrons are not enough for completely reducing the Pt precursors and more $PtO_2$ species were detected according to the XPS data; in addition, this $e^-$ deficiency was not enough for the oxygen vacancies generation producing a lower transition to the $Ti^{3+}$/$Ti^{4+}$ mixed phase ($NaTi_8O_{13}$) formation.

In the case of the NTO/Pt/Ni sample, a Pt-loaded protonated phase mixed with a lower quantity of $NaTi_8O_{13}$ is used as starting material. As it was explained before, during the Ni impregnation it is possible the $Na^+$ exchanges with $Ni^{2+}$ into the $TiO_6$ layers; in this context, as a result of the Na loss during this process it is able to grow the Na-deficient phase ($NaTi_8O_{13}$) thanks to the posterior thermal treatment.

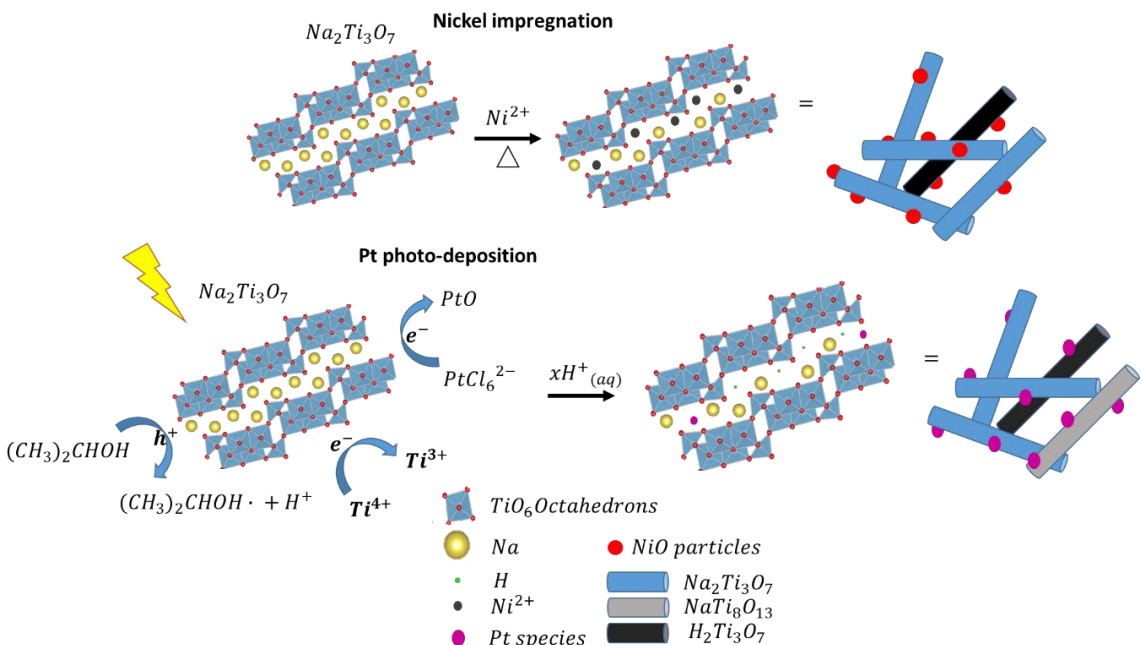

**Figure 8.** Proposed reaction mechanism of the phase transition during the $Na_2Ti_3O_7$ single Ni impregnation and Pt photo-deposition.

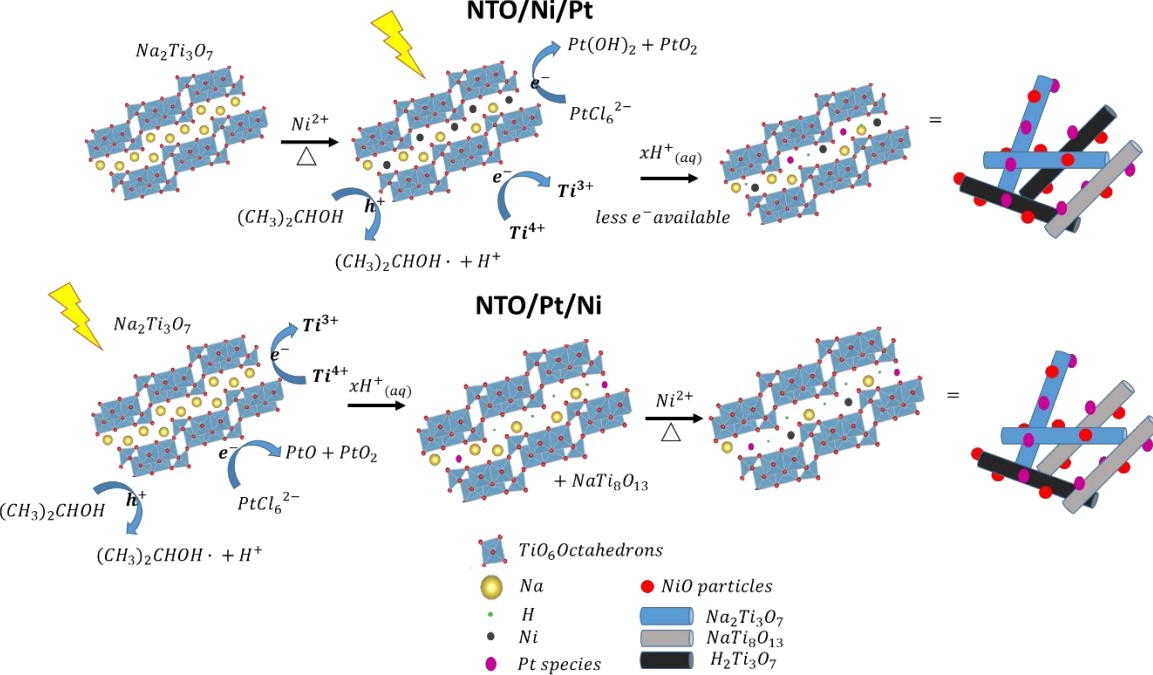

**Figure 9.** Proposed reaction mechanism of the phase transition during the bi-metallic $Na_2Ti_3O_7$ synthesis.

### 2.2. Photocatalytic Evaluation

Figure 10 summarizes the hydrogen evolution reaction rate of the experiments carried out under UV irradiation ($\lambda_{max}$ = 365 nm) in a continuous gas phase reactor. It can be seen that the bare sample (NTO) has a low hydrogen production reaction rate under these conditions (13 $\mu$mol$\cdot$g$^{-1}\cdot$h$^{-1}$). Hydrogen was not formed when the reaction experiment was carried out with bare glass substrates.

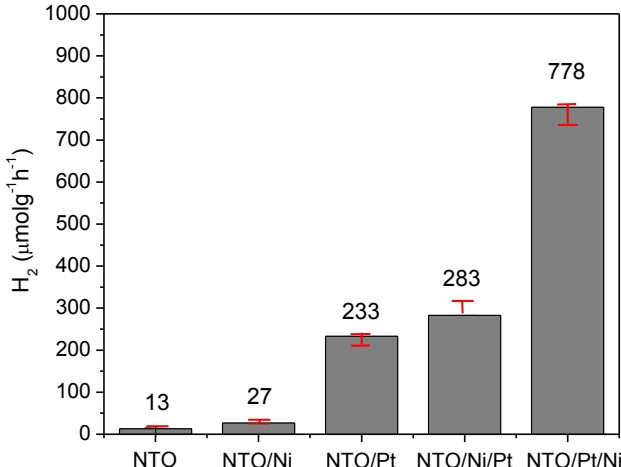

**Figure 10.** H$_2$ evolution rates from photocatalytic photo-reforming reaction under metallic and bi-metallic NTO samples.

The use of a single co-catalyst (Ni or Pt) on this NTO material leads to an increase in the hydrogen production, moderate in the case of nickel and higher in the case of the Pt, as shown in the same figure. This behavior can be mainly due to the presence of Pt on the NTO surface, which improves the carrier transfer step. Comparison on the effect of Pt and Ni deposition shows more than one order of magnitude improvement (factor 18) for Pt, while Ni deposition hardly favors the hydrogen evolution in relation to the bare NTO, with a two-fold increase [26,48].

The effect of the order in the metal deposition on the photo-reforming reaction efficiency is also clearly observed. As shown in Figure 10, the NTO/Ni/Pt sample (Ni impregnation before Pt photo-deposition) exhibits lower hydrogen reaction rate (283 vs. 778 $\mu$mol$\cdot$g$^{-1}\cdot$h$^{-1}$) compared to NTO/Pt/Ni sample (Pt deposition before Ni). This fact can be explained through the combination of several factors. First at all, as we have Pt deposited onto the NTO surface, which allowed the formation of the more reactive H$_2$Ti$_3$O$_7$ or Na$_{2-x}$H$_x$Ti$_3$O$_7$ phases [49], as demonstrated by the XRD, XPS, and EDS analysis. Secondly, the NTO/Pt/Ni catalyst has a higher surface area and pore volume than the other samples, implying a higher number of active sites for adsorption of ethanol-water molecules. Third, according to XPS, calculated Ti$^{3+}$/Ti$^{4+}$ ratios of the NTO/Ni/Pt catalyst present a higher quantity of oxygen vacancies which also correspond to reactive sites for the reaction [50]. Fourth, NTO/Pt/Ni apparently presents more Pt$^{2+}$ species than Pt$^{4+}$ ones, the Pt being the reduced species that is more efficient for photocatalytic hydrogen production [51–53].

The time-dependent rate of hydrogen evolution over the most efficient semiconductor catalyst (NTO/Pt/Ni) is presented in Figure S4. As seen, hydrogen production increases continuously during the first 5 h of reaction and remains stable over time (16 h), which is indicative of initial pre-activation and further catalyst stability. In addition, products of the oxidation reactions of ethanol were not detected during the experiments. These species may be adsorbed on the surface of the catalysts or they may be present in very low concentrations and could not be detected during the GC analysis.

Furthermore, in order to corroborate the catalyst stability, XPS and SEM characterization was performed on the recovered sample after the photocatalytic reaction experiments. The XPS results are shown in Figure S5. As seen, the formation of this Ni 2p spectra did not present significant variations in the position of peaks, which evidences the apparent stability in the oxidation state of Ni species

impregnated over titanate surface. In case of Pt spectrum, no peaks were practically detected due to the high intensity of C 1s species present in the surface, masking the low Pt 4f signals as well as those of the other buried layers. This increase in intensity in the C 1s spectrum can be related to the adsorption in the surface of some products not evolved as a result of the oxidative process.

Additional EDS mapping of the post-reaction NTO/Pt/Ni sample in Figure S6 shows the mixed and individual maps of Na, Ti, Ni, and O, clearly indicating that the whiskers retain the Na, Ti, and O composition, with a very homogeneous distribution of Ni.

## 3. Materials and Methods

### 3.1. Catalysts Synthesis

$Na_2Ti_3O_7$ synthesis (NTO) was performed by the solid-state reaction; for this purpose, stoichiometric proportions of anhydrous $Na_2CO_3$ (99% DEQ Químicos, Monterrey, México) and $TiO_2$ (Evonik P25, Essen, Germany) were perfectly mixed in an agate mortar using acetone as dispersant media. The wet slurry was transferred into a platinum crucible and thermally treated in air at 800 °C for 12 h.

Nickel oxide was deposited on the surface of the sodium titanate (NTO/Ni) by the wet impregnation method. The appropriate amount of nickel acetate (Fermont, Monterrey, México), calculated for 1% in weight, was dissolved in 40 mL of anhydrous ethanol. Then, $Na_2Ti_3O_7$ was added to the solution and mixed under vigorous agitation for 2 h. Then, the solvent was evaporated by heating at 110 °C. The dry solid was thermally treated at 400 °C for 2 h in air in order to transform the nickel organic salt deposited on the surface of the support to nickel oxide species.

Platinum was deposited by using a photo-deposition method (NTO/Pt) [54]. For this purpose, chloroplatinic acid hydrate ($\geq$99.9%, Sigma Aldrich, Darmstadt, Germany), calculated for 1% in weight of Pt, was dissolved in a water-isopropanol (0.3 M) mixture. $Na_2Ti_3O_7$ was added to the solution and the suspension was sonicated for 15 min. Photo-deposition was carried out by using an Osram lamp (300 W, I = 100 mW·cm$^{-2}$) under a continuous $N_2$ flow (0.5 mL·min$^{-1}$) for 2 h. Under this reaction conditions, only metallic platinum is expected to be deposited on the surface of the support. Finally, the suspensions were centrifuged and washed with Milli-Q water in order to eliminate remaining chlorine species. The wet solids were dried at 110 °C.

Bi-metallic catalysts were prepared by a similar procedure. The first batch of catalysts was labeled NTO/Ni/Pt indicating that nickel oxide species form the first layer of co-catalyst and the metallic platinum form the second layer. The second batch of bi-metallic catalyst was labeled NTO/Pt/Ni indicating that the metals were deposited in the reverse order.

### 3.2. Characterization

The structural characterization was performed with a Bruker D8 Advance X-Ray (Bruker Corporation, Billerica, MA, USA) diffractometer (CuK$\alpha$ radiation) equipped with an LYNXEYE super speed detector and a Ni filter over the 2θ collection range of 10–70° with a scan rate of 0.05° s$^{-1}$.

The morphology of the photocatalysts was analyzed with a Field Emission Scanning Electron Microscope (FESEM, Zeiss Auriga, Madrid, Spain) equipped with an Electron Dispersive X-Ray analyzer. XPS analyses were carried out in PHI 5500 Multitechnique equipment (Physical Electronics, Chanhassen, MN, USA) with Al K$\alpha$ radiation.

Diffuse reflectance UV–VIS spectra of all the photocatalysts were obtained in a UV–VIS NIR spectrophotometer (Cary 5000, Agilent Technologies, Santa Clara, CA, USA) coupled with an integrating sphere. The band gap of the samples was calculated with the Kubelka–Munk function.

BET surface area measurements were performed by $N_2$ adsorption–desorption isotherms using a Micromeritics TriStar II instrument (Micrometrics Instrument Corp, Norcross, GA, USA).

### 3.3. Photocatalytic Reactions

In order to evaluate the photocatalytic activity of the $Na_2Ti_3O_7$ and $Ni-Pt-Na_2Ti_3O_7$ catalysts for hydrogen evolution through ethanol reforming, several gas-phase experiments were carried out in a continuous reactor.

Catalysts were deposited on glass substrates previously cleaned sonicated with a mixture of acetone, isopropanol, and water during 15 min. Then, 0.1 g of powder catalysts were suspended in 6 mL of Mili-Q water and sonicated 15 min to form a homogenous slurry that was deposited by drop-casting over the glass substrates and naturally dried at room temperature for about 12 h. The remaining moisture was eliminated by heating at 80 °C during 15 min under air flow.

The glass substrates are placed inside a circular reaction cell illuminated with a Hamamatsu Lightningcure Spot LC8 UV-lamp (I = 21.8 mW·cm$^{-2}$ at 365 nm, Hamamatsu Photonics K. K., Bridgewater, NJ, USA) located at 15 cm above the quartz window. An argon stream (14 mL·min$^{-1}$) is bubbled through a water/ethanol mixture (50:50 *v/v*), kept at constant temperature of 35 °C, and continuously fed to the photocatalytic reactor.

The outlet of the reactor system is connected to a Varian 490-Micro GC (Agilent Technologies, Santa Clara, CA, USA) equipped with MS5A and PPQ columns. A sample of the gas stream was injected to the GC every 15 min to determine $H_2$ concentration.

A schematic representation of the reaction system is presented in Figure 11.

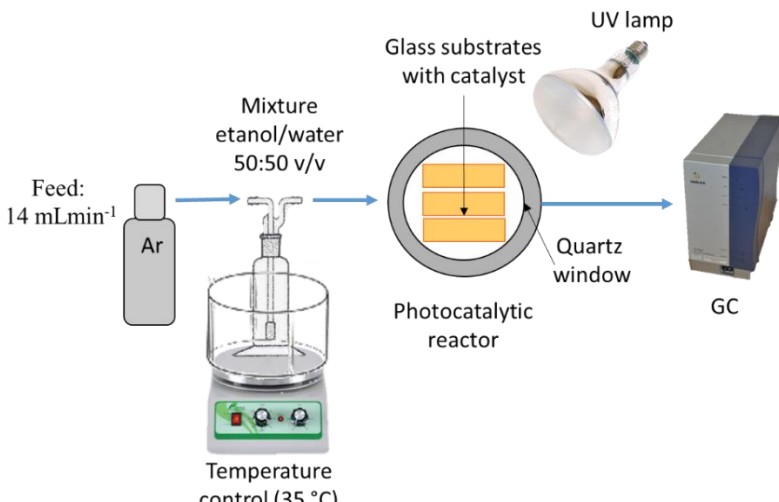

**Figure 11.** Schematic representation of the photocatalytic gas phase system used.

## 4. Conclusions

$Na_2Ti_3O_7$ whiskers were successfully synthesized by the solid state reaction. Ni and Pt were deposited over the titanate surface with different methodologies (wet impregnation for Ni and photo-deposition for Pt) resulting in the formation of NiO, $Ni(OH)_2$, and Pt (in different oxidation states, $Pt^{2+}$ or $Pt^{4+}$). Both metals gave raise to NTO modification observed in the XRD spectra.

Whereas Ni impregnation did not lead to a relevant increase in catalytic activity, Pt did, achieving an increment higher than one order of magnitude. The combination of both catalysts had a synergic effect, and its order of deposition was a very important factor. The previous nickel impregnation (NTO/Ni/Pt) did not lead to a significant increase in photocatalytic activity; on the contrary, the previous photo-deposition of platinum (NTO/Pt/Ni) led to almost three-fold higher productivity in the NTO/Ni/Pt sample (i.e., 778 versus 283 µmol·g$^{-1}$·h$^{-1}$, respectively). Aside from this difference in the photocatalytic performance, there are also other complementary differences between these NTO/Pt/Ni and NTO/Ni/Pt samples. First, in both cases, the samples present modifications in their XRD spectra, likely attributed to the formation of the reduced phase of the NTO. The experimental data evidence that the longer bi-metallic deposition times give rise to a higher partial reduction of

bare material creating phases with a mixture of $Ti^{4+}/Ti^{3+}$ valences and leading to the formation of protonated $H_2Ti_3O_7$ phases that origin these extra peaks in the XRD spectra. Second, the sample ending with nickel shows a higher surface area and larger porous volume pointing out a larger effective surface to interact with the alcohol molecules. Third, the oxidation state of the platinum is mainly $Pt^{2+}$ in the NTO/Pt/Ni sample whereas it is predominantly $Pt^{4+}$ in the NTO/Ni/Pt one. In all the case, nickel corresponds to $Ni^{2+}$ oxidation state related to the presence of NiO and/or $Ni(OH)_2$. All these features facilitate an increase in photocatalytic activity for the hydrogen evolution proving the synergic effect in the combination of these two catalyst pointing out the more significant role played by the platinum in direct contact with the NTO substrate and the complementary role played by the nickel and its higher effective area to facilitate the final charge transfer to the water-ethanol molecules.

**Supplementary Materials:** The following are available online at http://www.mdpi.com/2073-4344/9/3/285/s1, Figure S1: Close view of (0 0 1) main reflection on the prepared catalysts, Figure S2: XPS survey spectra of NTO, NTO/Ni/Pt and NTO/Pt/Ni samples, Figure S3: Kubelka–Munk spectra of $Na_2Ti_3O_7$ bare and deposited with Ni and Pt metals, Figure S4: Time dependent hydrogen evolution over NTO/Pt/Ni sample, Figure S5: XPS characterization of the sample with the best performance (NTO/Pt/Ni) post reaction, Figure S6: EDX mapping analysis of the NTO/Pt/Ni sample after photocatalytic test, Table S1: Semi-quantitative analysis of XPS peaks of bi-metallic samples.

**Author Contributions:** Methodology: L.F.G.-R. and S.M.-L.; supervision: T.A., L.M.T.-M. and J.R.M.; validation: S.M.-L.; writing–original draft: L.F.G.-R. and S.M.-L.; writing–review and editing: T.A., E.M., L.M.T.-M. and J.R.M.

**Funding:** This research was founded by Generalitat de Catalunya through the CERCA Program and M2E (2017SGR1246); CONACYT through projects CB-2014-237049, PDCPN-2015-487, Ph. D. scholarship 635249, and "Becas Mixtas 2017 Movilidad en el extranjero 291212"; SEP through PROFIDESS-PRODEP-25292. IREC also acknowledges support by the European Regional Development Funds (ERDF, FEDER) and by MINECO project ENE2017-85087-C3-2-R. S.M.-L. thanks European Union's Horizon 2020 and the Agency for Business Competitiveness of the Government of Catalonia for funding under the Marie Sklodowska-Curie grant agreement no. 712939 (TECNIOspring PLUS).

**Acknowledgments:** Authors thank to David Avellaneda from FIME-UANL for his valuable help with the XPS measurements.

**Conflicts of Interest:** The authors declare no conflict of interest.

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
