# Peer review of "Photocatalytic Hydrogen Evolution Using Bi-Metallic (Ni/Pt) Na2Ti3O7 Whiskers: Effect of the Deposition Order"

_catalysts, doi:10.3390/catal9030285_

Round 1
Reviewer 1 Report
Using sodium titanate wiskers, the authors have studied the effect of Pt and Ni impregnation on photochemical dehydrogenation of ethanol.. They find that mixed addition of Pt, then Ni, gives the best result. One thing I miss is what becomes of the ethanol. Also, especially because I did not manage to get access to the Supplement, it would be nice with a description of the apparatus (S 6) in the proper paper.
The higher activity of bimetal coated whiskers with platinum added first is interesting and is explained by higher content of Pt(II) relative to Pt(IV) which seems reasoable because C-H activation should be more efficient with Pt(II). Perhaps the authors could add some reference here.
Finally, the English needs some editing. A few examples: Line 18: Whereas wet impregnation with Ni gives only a slight increase in the activity, photodeposition of Pt increased the H2 production by more than one order of magnitude. Line 22: is carried out first, then Ni (NTO/Pt/Ni compared to 283....with the reverse order(NTO/Ni/Pt). (delete the rest of the sentence). Line 182: gives...sample because of (thanks to). Line 206 can be noticed.. Line 208: suggests again that facile phase transformation takes place only under... Line 354 later use.
This is only a few examples I am afraid.
Author Response
1. Using sodium titanate wiskers, the authors have studied the effect of Pt and Ni impregnation on photochemical dehydrogenation of ethanol. They find that mixed addition of Pt, then Ni, gives the best result. One thing I miss is what becomes of the ethanol. Also, especially because I did not manage to get access to the Supplement, it would be nice with a description of the apparatus (S 6) in the proper paper.
R. We moved Figure S6 to the manuscript; now Figure 11. You can find it on page 17, line 443.
2. The higher activity of bimetal coated whiskers with platinum added first is interesting and is explained by higher content of Pt(II) relative to Pt(IV) which seems reasoable because C-H activation should be more efficient with Pt(II). Perhaps the authors could add some reference here.
R. Thank you. Reference 51 (added in the first reviewed manuscript) explains the effect of Pt species in photocatalytic hydrogen evolution using TiO2-water. In this case, authors explained that the more active catalyst was the sample with the higher concentration of Pt0 compared vs samples with PtO2 (Pt4+), which is quite similar to our report (more reduced samples vs more oxide).
In addition, references 52 and 53 were added to the manuscript in order to reinforce this information. In both cases, authors report higher activities for photocatalytic hydrogen evolution when samples were loaded with higher quantities of more reduced species pf Pt as a result of the facility of charge transfer. You can find them cited on line 363, page 14.
3. Finally, the English needs some editing. A few examples: Line 18: Whereas wet impregnation with Ni gives only a slight increase in the activity, photodeposition of Pt increased the H2 production by more than one order of magnitude. Line 22: is carried out first, then Ni (NTO/Pt/Ni compared to 283....with the reverse order(NTO/Ni/Pt). (delete the rest of the sentence). Line 182: gives...sample because of (thanks to). Line 206 can be noticed… Line 208: suggests again that facile phase transformation takes place only under... Line 354 later use.
This is only a few examples I am afraid.
R. Thank you very much. We made the suggested corrections and a complete English review of the manuscript.
Reviewer 2 Report
This manuscript deals with ethanol photo-reforming over Na-titanate whiskers in combination with Pt-Ni metals. The authors investigated in particular the effect of the preparation method and concluded that the best results were obtained when the photodeposition of Pt is carried out previously to that of the Ni.
In my opinion, the paper contains some interesting information on this hot topic, but there are some disturbing key points, which require to be carefully addressed by the authors.
First of all the preparation methods used. These methods must be better specified and discussed. In fact there is some confusion in the description of the methods. Nickel was impregnated or deposited ? Pt was photodeposited or irradiated after impregnation ? What is referencia in line 317 ?.
Have the authors compared results with more conventional preparation approaches. I think this is really important to strengthen to the paper. In this regard it is also important to specify why photo-deposition of Pt was used ? This technique should provide metal catalysts (Pt° in our case), as also stated by the authors in line 322. However in the samples of this work the oxidized species (Pt2+ and Pt4+) are instead predominant, as shown by XPS. This should be deeply explained and discussed also in relation to the pre-treatment conditions used. These conditions are not clear in the paper.
Second the metal charge of samples. Which is the real concentration of Pt on the samples ? This should be measured with accuracy, for instance by ICP, to rule out that the different activity of samples can be related to the metal concentration on samples. The authors should also explain why quantification by EDS analysis (Table 2) provides very low Pt% values. What happened to Pt during the preparation steps?
Third the characterization of samples. The discussion of XRD patterns of Fig. 1 appears too simple. In fact the presence of more extra peaks in the NTO/Pt/Ni sample (lines 107-111 and Fig.1b) is also accompanied by some extra peaks present in the NTO/Ni/Pt sample (2theta 14, 24, 28 and 48 ca), which are not discussed. How are they justified ?
The discussion of Fig. 6 (DR spectra) is debatable, in so as Pt also reflects in the 500-600 nm region. How the lowest Eg of the NTO/Ni sample was explained ? Why this does not result in a higher photo-activity ? A comment on this point must be added.
XPS data: how samples were pretreated before the analysis? This must be reported considering that the oxidation of Pt can be occur on the surface. In line 185 the authors suggest a higher efficiency in Pt reduction over bare NTO than in Ni loaded NTO. Why not a lower re-oxidation rate ?
Forth, the photocatalytic results.
I find quite strange that no oxidation products (produced from ethanol) were detected with so long reaction times. It is hard to believe that adsorption on so low surface area samples can explain the lack of partial oxidation products. Please comment on this.
The reproducibility of experiments must be also indicated (for instance with bars in Fig. 7).
Minor points:
- I suggest the authors to avoid to report a summary of results in the introduction chapter (last paragraph)
- Lines 269-270: the increase of activity for Pt cannot be exactly defined as slight.
- Line 276: Fig. 7 instead of Fig. 6
- Line 283: Ti3+ instead of T3+
- Line 386: reasonably substrate instead of subtract
Therefore I think that the manuscript requires major revisions before being accepted for publication in Catalysts.
Author Response
This manuscript deals with ethanol photo-reforming over Na-titanate whiskers in combination with Pt-Ni metals. The authors investigated in particular the effect of the preparation method and concluded that the best results were obtained when the photodeposition of Pt is carried out previously to that of the Ni.
In my opinion, the paper contains some interesting information on this hot topic, but there are some disturbing key points, which require to be carefully addressed by the authors.
1. First of all the preparation methods used. These methods must be better specified and discussed. In fact there is some confusion in the description of the methods. Nickel was impregnated or deposited? Pt was photodeposited or irradiated after impregnation? What is referencia in line 317?
R. We are sorry for the confusion. We have made some corrections in the full text for a better understanding. In case of Nickel, it was impregnated over NTO (mixing of precursors, drying and calcination at 400 °C), and Pt photo-deposited (mixing of precursors with irradiation under N2 atmosphere and then drying). For the bi-metallic samples, the followed procedure was the same but in a different order depending on the sample.
We are sorry, we write the word “referencia” on line 317, now line 397 because we would add a reference on that line and we forgot. Reference was added an the mistake corrected. Thank you
2. Have the authors compared results with more conventional preparation approaches. I think this is really important to strengthen to the paper. In this regard it is also important to specify why photo-deposition of Pt was used ? This technique should provide metal catalysts (Pt° in our case), as also stated by the authors in line 322. However in the samples of this work the oxidized species (Pt2+ and Pt4+) are instead predominant, as shown by XPS. This should be deeply explained and discussed also in relation to the pre-treatment conditions used. These conditions are not clear in the paper.
R. We appreciate the reviewer’s comment, as this is an important aspect to consider. First of all, we would like to indicate that photodeposition is an extended metal deposition technique in photocatalysis, with the particular advantage of being simple and carried out at mild conditions. Of course, as the reviewer indicates, metallic forms are more likely expected after this process, as consequence of the photocatalytic reduction of the metal precursor (from higher oxidation states). This process, however, depends on the specific experimental conditions (concentration, pH, irradiance, etc.) and on the intrinsic photocatalytic properties of the starting material. In our case, the incomplete reduction of Pt species can be understood from the relatively low performance of the starting NTO material. Despite this, it is also important to emphasize that obtaining partially oxidized species (in particular metals such as Ag or Pt) under this method is not surprising and, moreover, that these species have been also reported to act as co-catalysts for HER or other reduction reactions. Although different performances can be expected as consequence of the formation of different kind of junctions, Pt(IV) and Pt(II) can also be expected to trap and transfer electrons to redox species on the surface (see for instance, https://doi.org/10.1038/srep43445 and https://doi-org/10.1016/j.apsusc.2010.09.037). We have added a short comment on the manuscript and included the reference 39.
Regarding the possibility of comparing with other deposition methods, although we consider it would be a valid and interesting approach, the fact of having bi-metallic co-catalysts on a particularly sensitive photocatalyst (in terms of stability during the modification treatments) would imply carrying out a new study in order to firstly evaluate the individual metal deposition and, later, to analyze the effect on the second metal deposition. In this particular case, we prefer emphasizing on the effect of the deposition order, which we have found to have a significant effect on the final performance, under the two selected deposition methods.
3. Second the metal charge of samples. Which is the real concentration of Pt on the samples? This should be measured with accuracy, for instance by ICP, to rule out that the different activity of samples can be related to the metal concentration on samples. The authors should also explain why quantification by EDS analysis (Table 2) provides very low Pt% values. What happened to Pt during the preparation steps?
R. We are very sorry, actually we do not have available an ICP equipment and taking in consideration the short time for sending the answers it was impossible to do the measurements. However, we added the weight percentage of Na, Pt and Ni calculated by EDS (in the first version we only reported atomic percentage) and as it can be noticed, the Ni and Pt w% are closed to 1% (initial concentration). Compared to the Ni concentration, the Pt one is lower. This difference is directly related to the deposition conditions; for instance, after Pt photo-deposition, some washings are made in order to eliminate the residual chlorines, and some Pt can be losing during this process. Meanwhile, after Ni impregnation, no washings were performed and it is possible to keep the initial concentration.
4. Third the characterization of samples. The discussion of XRD patterns of Fig. 1 appears too simple. In fact the presence of more extra peaks in the NTO/Pt/Ni sample (lines 107-111 and Fig.1b) is also accompanied by some extra peaks present in the NTO/Ni/Pt sample (2theta 14, 24, 28 and 48 ca), which are not discussed. How are they justified ?
R. Thank you very much for the observation. We made a deeper discussion in the XRD section mainly about the NTO/Ni/Pt and NTO/Pt/Ni as it follows:
“More complex features were observed in the bimetallic samples as shown in the XRD patterns in Figure 1b. As it can be noticed, both samples present more extra reflections with higher intensities than the single metal loaded ones, which are representative of a modification of the pristine phases. They may be due to Na deficiency in the crystalline structure of sodium titanate and these reflections may indicate the formation of NaTi8O13 and H2Ti3O7 phases (JCPDS 00-048-0523 and 00-036-0654, respectively), being the first one representative of a family of sodium titanates with Ti3+/Ti4+ mixed-valence [30]. In addition, it is evident that the (0 0 1) reflection moved to slightly higher 2θ values, which also suggests the partial replacement sodium atoms to form Na2-xHxTi3O7. According to some studios, these phases are intermediate products of the transformation of Na2Ti3O7 to Na2Ti6O13 under extreme conditions [31]. In this context, we can assume that the bi-metallic deposition on the surface of sodium titanate under the performed reaction conditions favors the partial reduction of titanium and the substitution of the sodium atoms by hydrogen atoms.
It is evident that the intensity of the (0 0 1) reflection (≈ 10.5°) is different comparing bare and bi-metallic samples. The most noticeable decrease in this pattern is presented by NTO/Ni/Pt one; however, NTO/Pt/Ni is also less intense compared to the bare NTO (Figure S1). This reduction in the (0 0 1) titanate main intensity is related with a decrease in the interlayer distance between TiO6 octahedrons [32] suggesting a major replacement of Na by smaller cations such as Pt, Ni or H in the NTO/Ni/Pt sample and reducing the interlayer distance; furthermore, this sample presents more intense peaks of the main H2Ti3O7 phase (24,28 and 48°) compared to the NTO/Pt/Ni, corroborating this interlayer reduction and suggesting the major presence of this phase instead of the NaTi8O13. A contrary behavior is observed in NTO/Pt/Ni sample, where H2Ti3O7 phase peaks are less intense compared to the NaTi8O13 (17.8 and 27.8°), suggesting the higher presence of the Ti3+/Ti4+ mixed-valence on this catalyst.
A less marked behavior is observed in single metal loaded samples, as it is observed, also in Figure S1, Ni and Pt loaded NTO present a reduction in the (0 0 1) reflection intensity, as a result of the decrease in the interlayer distance, more evident in NTO/Ni sample, which suggests the possible introduction of Ni species between the TiO6 octahedrons.
In this context, it is important to highlight that the metal deposition order produces a different development in the formed faces from NTO”.
You can find this explanation in blue in the full text on page 3 from line 95-125.
5. The discussion of Fig. 6 (DR spectra) is debatable, in so as Pt also reflects in the 500-600 nm region. How the lowest Eg of the NTO/Ni sample was explained? Why this does not result in a higher photo-activity? A comment on this point must be added.
R. Thank you very much for the observation. Some of that transitions in Pt loaded samples (between 500-800) are representative of PtO2 or Pt(OH)2, species which were marked in the Figure 7. In addition, it was explained that the decrease in the bandgap of NTO/Ni sample can be related to the Na exchange by Ni between the TiO6 octahedrons as a result of the impregnation conditions. The full explanation was highlighted in blue in the full text on page 11, from line 271 to 292.
The bandgap reduction on NTO/Ni sample enhanced the photocatalytic activity if we compare to the bare NTO; however, as it was explained in the full text, the higher efficiency was related with some other phenomena such as the presence of oxygen vacancies, higher surface area and pore size distribution, the presence of more reduced Pt species (Pt2+), etc.
6. XPS data: how samples were pretreated before the analysis? This must be reported considering that the oxidation of Pt can be occur on the surface. In line 185 the authors suggest a higher efficiency in Pt reduction over bare NTO than in Ni loaded NTO. Why not a lower re-oxidation rate?
R. We understand the reviewers’ concerns on this matter. The XPS measurements were carried out without previous pre-cleaning or pre-treatment of the samples, as we did not want to modify the surface. As previously indicated, we believe the relative higher oxidation state of Pt species are related to incomplete reduction during photodeposition, rather than to experimental modification during XPS analysis. The fact of having less oxidized Pt on the Ni-loaded NTO can be explained through a possible change on the deposition process caused by the presence of Ni2+ species on the surface, namely, modification of the charge transfer dynamics, through a Ni2+/Ni0 redox process or even through a favored HER (instead of Pt (IV) reduction). It is important, however, to highlight that despite the relative lower Pt2+/Pt4+ ratio, the total amount of Pt seems to be higher on this sample, as indicated in Table 2, which ultimately agrees with the expected improvement on the photodeposition as consequence of the preliminary deposition of Ni.
Forth, the photocatalytic results.
7. I find quite strange that no oxidation products (produced from ethanol) were detected with so long reaction times. It is hard to believe that adsorption on so low surface area samples can explain the lack of partial oxidation products. Please comment on this.
R. Once again, thank you for the comment. It is true that some partial or total oxidation products were expected. However, as confirmed by the XPS measurement on the NTO/Pt/Ni sample after a long photocatalytic test, the significant presence of adsorbed C-O and C=O suggests that most of the oxidation products could remain on the surface, forming carbonates or other intermediate products. In fact, probably the amount of oxidized species is very low, considering also that besides acting as HER catalyst, Ni species (in the form of oxides or hydroxides) are also able to trap holes, thus intrinsically depleting the oxidation of the organic compounds (i.e. ethanol).
8. The reproducibility of experiments must be also indicated (for instance with bars in Fig. 7).
R. Thank you very much, we added some bars on figure 7
9. Minor points:
- I suggest the authors to avoid to report a summary of results in the introduction chapter (last paragraph)
R. We eliminated the summary in the introduction
- Lines 269-270: the increase of activity for Pt cannot be exactly defined as slight.
- Line 276: Fig. 7 instead of Fig. 6
- Line 283: Ti3+ instead of T3+
- Line 386: reasonably substrate instead of subtract
R. We did the appropriate corrections in all cases.
Therefore I think that the manuscript requires major revisions before being accepted for publication in Catalysts.
Reviewer 3 Report
The paper needs an extensive English editing.
I have 2 main observation to improve the quality of the presentation:
1 - page3 : lines 103-111. The authors should provide a reaction scheme that would be helpful to understand what are the products formed and how these form. I think that that the UV illumination leads to the formation of oxygen vacancies with formation of H2Ti3O7.
2 - Figure 2, 3 and discussion of XPS data: The authors should include the Ti2p signal from the starting material NTO. Also the Ti 2p fitting is not too convincing. In my opinion the amount of Ti3+ is too high and this is the reason why a comparison with the starting material would be helpful.
Author Response
The paper needs an extensive English editing.
I have 2 main observation to improve the quality of the presentation:
1 - Page3: lines 103-111. The authors should provide a reaction scheme that would be helpful to understand what are the products formed and how these form. I think that that the UV illumination leads to the formation of oxygen vacancies with formation of H2Ti3O7.
R. Thank you very much for your comment. Two diagrams with their respective explanation were added to the full text in order to explain what happens during the deposition of a single and both metals. You can find Figures 8 and 9 on pages 12 and 13, and their respective explanation in blue from lines 302 to 336.
2 - Figure 2, 3 and discussion of XPS data: The authors should include the Ti2p signal from the starting material NTO. Also the Ti 2p fitting is not too convincing. In my opinion the amount of Ti3+ is too high and this is the reason why a comparison with the starting material would be helpful.
R. XPS analysis were performed to the bare NTO sample, you can find the Na 1s, Ti2p and O 1s spectra on figure 2 and its respective explanation in blue from line 137 to 143. In case of the Ti 2p spectrum, we did not find Ti3+ species as a result of its de-convolution.
Round 2
Reviewer 2 Report
The authors addressed all the questions arisen.
In particular they specified better the preparation methods, enlarged the discussion of the results, adjusted the figures and corrected some typographical errors.
Therefore the paper was strongly improved and I now recommend its publication in Catalysts